# High-Efficiency Extraction of *Pantoea alhagi* Exopolysaccharides Driven by pH-Related Changes in the Envelope Structure

**DOI:** 10.3390/molecules27217209

**Published:** 2022-10-25

**Authors:** Yuhang Ma, Liang Sun, Rui Wang, Yian Gu, Hong Xu, Peng Lei

**Affiliations:** State Key Laboratory of Materials-Oriented Chemical Engineering, College of Food Science and Light Industry, Nanjing Tech University, Nanjing 211816, China

**Keywords:** exopolysaccharides, extraction, pH, rheological properties, *Pantoea alhagi*

## Abstract

Increasing numbers of exopolysaccharides and their properties have been explored. However, the difficulty of extracting high-viscosity exopolysaccharides has hindered their further industrialization. In this research, we explored a strategy based on encapsulated structure control under different pH to efficiently extract *Pantoea alhagi* exopolysaccharides (PAPS). Results showed that at pH levels of 6, 12, and 13, the extraction efficiency of PAPS was elevated, and the yield did not decrease. The rheological properties of the pH−12-treated PAPS were better than those of PAPS treated at pH 7, while the pH−6-treated PAPS decreased. The effects of pH−12-treated PAPS on soil macroaggregates and soil’s water evaporation rate were similar to those of PAPS treated at pH 7. In addition, we observed that treatment at pH 12 produced a significantly reduced encapsulated structure compared with treatment at pH 7. The proportion of unsaturated fatty acids after treatment at pH 12 was higher than after treatment at pH 7, which may result in reduced encapsulated structure in pH−12 conditions. These results enrich the understanding of the effect that alters pH conditions on the encapsulated structure to improve the extraction efficiency of exopolysaccharides and provide a theoretical basis for the extraction of exopolysaccharides with extreme viscosity.

## 1. Introduction

Microbial polysaccharides consist of endopolysaccharides and exopolysaccharides, which have great potential for application in the fields of food, biomedicine, cosmetics, and agriculture [1,2,3,4]. Exopolysaccharides are usually fermented and extracted from the extracellular surface by microorganisms, including bacteria and fungi. Compared with polysaccharides derived from plants and animals, exopolysaccharides have more stable molecular weight, better yield, a shorter production period, and lower cost due to their stable fermentation process [5]. Moreover, most bacterial exopolysaccharides are more stable in production than fungal exopolysaccharides. It was reported that *Ganoderma lucidum* production of polysaccharides is directly affected by the morphology of mycelial pellet. Small, loosely branched mycelium pellets produce higher polysaccharides compared to large mycelium pellets [6]. Therefore, more and more microbial exopolysaccharides, such as xanthan, levan, dextran, pullulan, alginate, and hyaluronic acid, are commercially produced in a laboratory [5,7,8].

PAPS, derived from *Pantoea alhagi* NX-11, is a high-molecular-weight anionic biological macromolecule which has been shown to play an important role in root colonization and in enhancing crop stress tolerance [9,10]. Furthermore, it was found that exogenous addition of PAPS enhanced the drought stress resistance of rice seedlings [11]. Subsequently, by increasing the dissolved oxygen level in the highly viscous fermentation system, the PAPS yield was increased to 19.27 g/L, which greatly increased its potential for industrial application. Unfortunately, viscosity of the fermentation broth used to produce the higher yield exponentially increased the difficulty of downstream separation of PAPS.

After fermentation, downstream processing is a key step to ensure a high recovery rate of refined-grade exopolysaccharides [7,12]. This stage may affect key exopolysaccharide properties such as molecular weight, yield, monosaccharide composition, and physicochemical properties of the polymer, and crucially, it is the most costly part of actual production [13]. For viscous exopolysaccharides with high molecular weights, the conventional means of separation is usually to inactivate microorganisms and some undesirable enzymes, such as glucanase. Subsequently, the broth is diluted 3–10 times and the bacteria removed by centrifugation or filtration; then, it is concentrated to 1/5–1/3 of the original volume. The crude extract is obtained by alcohol precipitation [13,14,15]. In this process, the dilution and concentration consume a great deal of energy and time, which is clearly impractical for industrial production. In order to solve this problem, researchers have tried many approaches, such as subzero extraction, enzyme-assisted extraction, ultrasonic-assisted extraction, microwave-assisted extraction, and ultrasonic-microwave synergistic extraction [15,16,17,18,19]. Although these methods have different advantages in terms of extraction efficiency and impact on polysaccharide function, industrial-scale extraction is still a long way off. Therefore, depending on the desired purity or intended use, the optimization of this processing stage may be imperative.

Exopolysaccharides and bacteria form an encapsulated structure, which is one of the reasons why the exopolysaccharides are difficult to extract. Nancy et al. reported that *Sphingomonas*-secreted exopolysaccharides adhered to the bacterial surface to form an encapsulated structure, which reduced the mass transfer rate and severely limited the improvement of production efficiency [20]. More recently, Wang et al. achieved production of high-yield hyaluronic acid and hyaluronic acid of different molecular weights by disrupting the encapsulation of leech hyaluronidase [21]. Zhao et al. successfully altered the capsule of *Sphingosphinomonas* strains by knocking out the sortase gene (srtW) and obtained welan gum with low molecular weight [22]. Therefore, removing the encapsulated structure is one potential way to improve the extraction yield.

In this study, we tried to screen out a simple separation process through the regulation of pH. At the same time, we measured the yield, molecular weight, monosaccharide composition, rheological properties, and influence on the soil structure of PAPS, observed the microstructure of PAPS and *P. alhagi* NX-11, and analyzed the fatty acid composition of cell membranes treated under different pH conditions, so as to comprehensively determine the effect of different envelope structure changes on PAPS extraction at different pH levels. We hope that this study can provide a reference for the efficient industrial extraction of polysaccharides from high-viscosity systems.

## 2. Results

### 2.1. Effects of Different pH on the Extraction Process and PAPS Yield

The picture of dried PAPS has been provided in Appendix A and the picture of *P. alhagi* NX-11 in Appendix A. Figure 1 shows the state of the PAPS extraction process under different pH conditions and the effect of pH on PAPS yield. Under neutral pH conditions, the yield of PAPS was 19.28 g/L (Figure 1D), the bacteria were separated by centrifugation at 12,500× *g*, 10 min for eight times (Figure 1B), and the 10 g/L PAPS solution exhibited a gel shape (Figure 1C). With low pH (pH 1–4), the supernatant was separated from the bacteria only by centrifugation at 12,500× *g*, 10 min for one time (Figure 1B), and the PAPS solution was in a liquid state (Figure 1C). However, the yield of PAPS was significantly reduced (Figure 1D). The yield of PAPS extracted at pH 1–2 was reduced to less than 10 g/L. This change could be due to the high concentration of acid under heating conditions causing PAPS hydrolysis [23], thus reducing the yield of PAPS and the viscosity of the fermentation broth. When the pH of the fermentation broth was adjusted to 5, the PAPS yield significantly decreased (*p* = 0.0215), while no change in PAPS yield was observed at pH 6 (Figure 1D). Despite the difference in yield, the fermentation broth and PAPS solutions with pH levels of 5 and 6 showed similar phenotypes (Figure 1B,C). When the pH of the fermentation broth was adjusted to alkaline, the phenotype showed more differences. When the pH was in the range of 8–10, for example, the fermentation broth needed to undergo centrifugation 10 times to separate the bacteria and obtain the supernatant. This could be because the addition of sodium ions improved the viscosity of the fermentation broth, while the influence of the weak alkali on the molecular weight and composition of polysaccharides was small [24,25]. The process of multiple high-speed centrifugations greatly increased the time and energy consumption of PAPS extraction, making it impractical for commercial application. When the pH of the fermentation broth was adjusted to 12, the clarified supernatant could be obtained by centrifugation two times. At the same time, the yields of PAPS extracted at pH 8–12 were not significantly different from that extracted at pH 7, and the PAPS solution was similar to that at pH 7, presenting a gelatinous appearance. However, when the pH was further raised to 13 or 14, there were some interesting differences. The color of the fermentation broth was darker, and the centrifugation efficiency was higher (centrifuged for once) (Figure 1). At the same time, the PAPS solution also showed better fluidity (Figure 1C). This could be because the partial hydrolysis of PAPS under acid or base conditions led to the decrease of the extraction yield of PAPS. Surprisingly, the yield of PAPS treated with pH 13 was not significantly different from that obtained under normal conditions, which was quite different from the results for pH 1–4, but the yield of PAPS treated with pH 14 was significantly reduced (*p* = 0.0019).

### 2.2. Effects of Different pH on the Molecular Weight and Monosaccharide Composition of Crude PAPS

It is well known that strong acids or bases reduce the molecular weight of PAPS [26]. However, the effect of different pH on the specific molecular weight of PAPS has not been reported yet. In this study, we used GPC to qualitatively analyze the molecular weight of PAPS under different pH conditions. As shown in Figure 2, the molecular weight of PAPS was not reduced by under pH 5–12 conditions. At low pH (pH 1–4), the molecular weight of PAPS was divided into two parts, and we suspect that this is resulted in partial acid hydrolysis of PAPS, both of which were significantly lower than that of PAPS extracted at pH 7. In addition, high-pH (pH 13–14) conditions significantly reduced the molecular weight of crude PAPS (Figure 2), which was closely linked to viscosity reduction of the fermentation broth and a fluid appearance of the PAPS solution (Figure 1B,C).

In order to investigate the differences between PAPS extracted at different pH values, we characterized the monosaccharide components. Figure 3 illustrates the differences in monosaccharide components at different pH levels. The monosaccharide components of all samples were composed of glucose, galactose, mannose, glucuronic acid, and glucosamine (Appendix A). Overall, the monosaccharide content was similar under the different pH conditions. For example, the PAPS contents of all samples were, in descending order, galactose, glucose, glucuronic acid, glucosamine, and mannose. There are some differences, though. Specifically, the percentage of galactose was lowest at pH 10 (36.29%) and highest at pH 7 (54.80%). The percentage of glucose was highest at pH 1 (31.58%), while the percentage of glucuronic acid was highest at pH 11 (18.62%). We noted that there was no correlation between differences of monosaccharide components and changes in pH. This is not the first time this phenomenon has been observed, though it is somewhat different from what might be expected. Wang et al. (2021) extracted polysaccharides from Actinidia deliciosa using different extraction methods and found that the differences in the monosaccharide fractions of the polysaccharides were significantly dependent on the extraction process [27]. A similar phenomenon was observed with the polysaccharide extracted from Symphytum officinale L. by Shang et al. [27].

### 2.3. Effects of Different pH on the FT-IR Analysis of Crude PAPS

According to the study by Ahmad Usuldin et al., the FT-IR spectra of PAPS samples treated with different pH were plotted as shown in Figure 4 [28]. There were many common characteristic absorption peaks among them, which means that they had many of the same organic functional groups [29]. The strong and broad typical peaks near 3410 cm^−1^ represented stretching vibrations of the hydroxyl group [30]. We attribute the peak at approximately 2930 cm^−1^ to a C−H stretching band in a sugar methylene group [31]. The absorption peaks at approximately 1620 cm^−1^ may represent the asymmetric C=O stretching vibrations of the carboxyl group, indicating the presence of uronic acids, and it may also be due to the bound water [32]. In addition, the absorption at approximately 1150 cm^−1^ indicated a pyranose, which was consistent with the results of monosaccharide composition analysis [33]. However, there were some different characteristic absorption peaks between the polysaccharide samples. Among these, the FT-IR spectra of polysaccharides treated with pH 1 were the most divergent from those of other conditions, which may be because some organic groups reacted under the action of a strong acid. In the pH−1-treated polysaccharide, the absorption peak around 1731 cm^−1^ was assigned to the C=O stretching vibration, whereas at pH 2–14, this characteristic absorption peak did not appear [30]. On the contrary, at pH 2–14, a characteristic absorption peak appeared near 1400 cm^−1^, which we attribute to deformation stabilization of C–H [34]. In addition, pH 2–14 samples all exhibited feature-absorption peaks around 1150 cm^−1^ and 1080 cm^−1^, which represent C−O−C and C−O−H retractive vibration [35]. Compared with pH−2–14-treated PAPS, there is another interesting phenomenon; the absorption peak of pH 1 near 3410 cm^−1^ was significantly lower than that of other solutions, which indicates that strong acid conditions affect both inter-molecular hydrogen bonds and intra-molecular hydrogen bonds [26]. Since NMR analysis is a heavy job, we stopped at FT-IR; NMR will be analyzed in subsequent studies.

### 2.4. Effects of Different pH on the Rheological Properties of Crude PAPS

The rheological properties of polysaccharides are the most intuitive embodiment of their physical properties, simply and clearly reflecting the differences in the physical properties of different polysaccharides and also pointing the way for the potential application of polysaccharides [14]. Therefore, we investigated the rheological properties of PAPS under different pH extraction conditions.

#### 2.4.1. Apparent Viscosity and Flow Curves

Regardless of the pH conditions of PAPS extraction, the viscosity of polysaccharides decreased with increased shear rate, and this trend was independent of molecular weight (Figure 5A,B), indicating that PAPS from *P. alhagi* NX-11 had a shear-thinning fluid behavior characteristic of non-Newtonian fluids with pseudoplastic character [36,37]. This pseudoplastic behavior may be caused by the gradual disentanglement and orientation of polysaccharide molecules in the shear flow field [38,39]. Obviously, the molecular weight of PAPS affected its viscosity. The viscosity of the PAPS treated with pH 1–4 and pH 13–14 was lower than that of PAPS treated with pH 5–12 between 0 and 1800 S^−1^ (Figure 5A,B). However, this difference was also apparent in the high-molecular-weight PAPS samples. The viscosity of pH−8–12-treated PAPS was significantly higher than that of PAPS treated at pH 5–7 (Figure 5A,B). We speculate that because Na^+^ forms an ionic bond with the uronic acids of PAPS, it was also precipitated along with the PAPS during alcohol precipitation and that the presence of Na^+^ significantly strengthened the intermolecular network structure, thus maintaining a high viscosity at a high shear rate [39,40]. In addition, in the PAPS samples treated at pH 1–6 and pH 13–14, the viscosity trended to zero when the shear rate was greater than 600 s^−1^, which may be because the inter-molecular tangles in these samples were not tight enough, and the higher shear rates untangled their molecular chains, thus rapidly reducing viscosity [41]. This viscosity data also confirmed the phenomenon we observed in the PAPS solution (Figure 1B). Moreover, the pseudoplastic fluid properties of PAPS indicated possible directions for its application in other fields, such as the food industry, adhesives, and organic materials [42].

The variations of shear stress with increasing shear rate for PAPS under different pH conditions are shown in Figure 5C,D. PAPS solution stress increased with increasing shear rate. PAPS treated at pH 1–4 and pH 13–14 were found to have significantly lower molecular weights (Figure 2), and it is likely that this reduction in molecular weight is responsible for the better dispersion of PAPS molecules in the aqueous solution and for weaker interactions. This shows that the shear stress and shear rate are nearly linearly related, and the PAPS solutions were similar to Newtonian fluid [43]. In contrast, the molecular weight of PAPS treated at pH 5–12 did not change significantly, and the relationship between shear stress and shear rate was completely different from that of PAPS treated at pH 1–4 or pH 13–14, presenting a nonlinear relationship. This may have been caused by blockage of the free movement space of molecules in the macro-molecular PAPS solution and covering and overlapping between molecules [37]. In addition, with higher degrees of PAPS polymerization, the probability of forming hydrogen bonds between molecules was greatly increased. However, due to the relatively small bond energy of the hydrogen bond, the inter-molecular hydrogen bond was destroyed to a certain extent under the action of the external shearing force, causing the PAPS solution treated at pH 5–12 to show the effect of shear thinning and giving it the properties of non-Newtonian fluid [44,45].

#### 2.4.2. Storage (G’) and Loss (G”) Moduli

Of the rheological properties of PAPS, the properties within the linear range of storage modulus (G’) and loss modulus (G”) best reflect the aggregation structure properties of substances. The G’ means that the stress energy can be recovered after being temporarily stored in the test, while the G” means that the irreversible loss is converted into heat. As can be seen from Figure 6, within the range of the oscillation frequency tested, the G’ and the G” increased with the increase of oscillation frequency, due to the fact that with increasing frequency, the chain segments do not have time to fully extend their orientation. The G’ values of the samples treated at pH 1–7 increased gradually with increasing pH. Samples treated at pH 8–12 always had a higher modulus of G’, and there was no significant relationship between them. The G’ of the PAPS treated at pH 13–14 decreased to a lower level. Similarly, the G” of PAPS in response to pH was in perfect agreement with the G’ (Figure 6B). In addition, we found that the G” at pH 1–6 and pH 13–14 was always higher than the G’ over the range of scanning frequencies, thus creating liquid-like behavior, while the G’ at pH 7–12 was always higher than the G”, thus leading to solid-like behavior (Appendix A) [40,46]. This feature was also visualized in Figure 6C, where there were two intersections of G” and G’ between pH 6–7 and pH 13–14, causing the samples to show different characteristics at different pH levels (Figure 6D).

### 2.5. Effects of PAPS on Water-Stable Soil Macro-Aggregates and Water Evaporation

Formation of water-stable soil macro-aggregates (>250 mm) is important for carbon sequestration, water retention, water movement, soil biodiversity, biological activity, plant growth, and a wide range of physical and biogeochemical processes [47,48,49]. It has been reported that secretion of PAPS could be the main reason why many plant promoters play a dominant role in the accumulation of water stability [50,51,52,53]. The effects of different pH-level conditions of PAPS on soil’s water-stable macro-aggregates and their effects on soil’s water-holding properties are shown in Figure 7. Apparently, the PAPS treated at pH 1–4 and pH 13–14 had reduced formation of water-stable soil macro-aggregates compared with untreated PAPS (pH 7). However, the PAPS treated at pH 5–12 showed no significant difference in the formation of water-stable soil macro-aggregates (Figure 7A). This suggested that the molecular weight of PAPS plays a crucial role in the formation of water-stable soil macro-aggregates, because the molecular weights at pH 1–4 and pH 13–14 were both decreased (Figure 2). This also indicated that the viscosity of the PAPS most likely does not have a significant effect on the formation of macro-aggregates, as the viscosity at pH 8–12 was significantly higher than that at pH 5–7 (Figure 7). In addition, Sheng et al. found that PAPS concentration was positively correlated with formation of water-stable soil macro-aggregates, while the source of PAPS did not seem to make a significant difference [53].

It is worth noting that the effect of PAPS on the soil’s water evaporation rate is the same as that of water-stable soil macro-aggregates (Figure 7). The high-molecular-weight PAPS (pH 5–12) effectively inhibited evaporation of soil water. At 25 ℃, the evaporation rate of soil water after 30 days of placement was about 40% (in the pH 5–12 groups), while the evaporation rate of water in the pH 1 and pH 2 groups reached as high as approximately 85% (Figure 7B). Similarly, there was no significant difference in water evaporation rate between pH 5–7 and pH 8–12, which shows that the viscosity of PAPS has no significant effect on soil water-retention ability.

PAPS extracted at pH 5–12 showed good effects on water-stable soil macro-aggregates and good water-retention ability, which may be a physical effect of increasing crop growth, as opposed to the biochemical effects in promoting crop resilience. Lin et al. (2016) proved that PAPS secreted by *Pseudomonas putida* X4 interacted with the minerals kaolinite, montmorillonite, and goethite through electrostatic interactions [54]. Moreover, it has been found that anionic polysaccharides can act as cation reservoirs, thereby reducing crop damage from heavy metal ions, sodium ions, etc., and enhancing abiotic stress resistance [55,56].

### 2.6. Comparison of Microstructures of PAPS and P. alhagi NX-11 Treated under Different pH Conditions

According to the study of Hanafiah et al., the microstructures of PAPS and *P. alhagi* NX-11 treated under different pH conditions were plotted as shown in Figure 8 and Figure 9 [57]. Based on the results of previous experiments, we selected several representative samples for analysis (pH 2, pH 7, pH 9, pH 12, and pH 13). The SEM images of PAPS are shown in Figure 8. The PAPS formed disordered random coils and exhibited sheet and tubular entangled structures, with circular clumps and chain formation suggesting that PAPS might form a cross-linked network. The PAPS extracted at pH 7–13 showed a combination of network, sheet, and tubular entanglement structures, and with increasing pH, the tightness of the network structure between polysaccharides decreased, possibly due to the decrease in molecular weight. The smooth sheet-like structure observed at pH 2 may be caused by a reduction of the interaction force between polysaccharides, due to the breaking of glycosidic bonds in polysaccharides [58]. These results are consistent with the molecular weight results.

As shown in Figure 9, the morphology of bacteria differed significantly under different pH conditions. At pH 7, bacteria were aggregated in the shape of obvious short rods; the bacteria were thick and blunt at both ends, and the bacterial surface was rough. The bacteria in the pH−9 solution were looser than in the pH−7 solution. At pH 7, there was an obvious envelope structure. Compared with bacteria in the pH−7 and pH−9 solutions, bacteria in the pH−12 solution had no strong envelope structure. Most *P. alhagi* bacteria without complete bacteria morphology or envelope structure of exopolysaccharides and bacteria were observed at pH 2 and pH 13. We speculate that this phenomenon may be caused by a disruption of bacterial integrity and hydrolysis of exopolysaccharides at extreme pH [23].

### 2.7. Fatty Acid Composition of Cell Membranes Treated at Different pH

In order to verify the relationship between extraction efficiency and cellular structure, we analyzed the membrane fatty acid composition under different processing conditions. The composition of fatty acids has an important influence on their permeability and fluidity, and changes in membrane composition and properties are an important factor in adaptation to different conditions. The U/S value of fatty acids in the pH−9 solution was significantly higher than that of fatty acids in the pH−7 solution (*p* < 0.05), and the proportion of oleic acid long-chain unsaturated fatty acids (25.30%) was significantly higher than in the pH−7 solution (*p* < 0.05). The U/S value of pH−12-treated fatty acids was significantly higher than that of pH−9-treated fatty acids (*p* < 0.05), and the proportion of oleic acid long-chain unsaturated fatty acids (29.43%) was significantly higher than in the pH−9 solution (*p* < 0.05) (Table 1). Increases in unsaturated fatty acids have been reported to enhance the permeability of *Lederaviaplasmic* B cells [59]; this was accomplished by treatment to alter the fatty acid composition of the cell membrane, which in turn affects PAPS separation from bacteria.

## 3. Materials and Methods

### 3.1. Materials and Chemicals

The strain with high PAPS yield (19.27 g/L), *Pantoea alhagi* NX-11, was previously isolated from sea-rice roots [8,10]. HCl, NaOH, standard monosaccharides, trifluoroacetic acid (TFA), and 1-phenyl-3-methyl5-pyrazolone (PMP) were purchased from Sigma Chemicals Co., Ltd. (St. Louis, MO, USA). The soil was collected from the campus of Nanjing Tech University (118.6422 E, 32.0754 N).

### 3.2. Fermentation in Shake flask

The fermentation methods refer to Sun et al. and Abdullah et al. [6,10]. The method used for fermentation preparation involving two culture stages with both stages cultivated at 30 °C with an initial pH of 7 and 200 rpm for 12 h and 24 h. Shake flasks of 500 mL were used to culture *P. alhagi* NX-11. The seed medium was LB medium, and the fermentation medium was compared with 40 g/L sucrose, 5 g/L NaCl, 4 g/L tyrptone, 2 g/L K_2_HPO_4_, 2.33 g/L n-hexane, 5.13 g/L n-heptane, and 9.06 g/L n-hexadecane, which were used to produce PAPS.

### 3.3. The PAPS Extraction Process under pH Conditions

The PAPS fermentation broth was obtained by fermentation of *P. alhagi* NX-11 with an initial pH of 7.0. The pH of the samples was adjusted to 1–14 by HCl or NaOH. Fourteen PAPS samples of pH 1–14 were heated (100 °C, 30 min) and magnetically stirred to fully contact and react. Subsequently, centrifugation was repeated at 10,000× *g* once every 10 min to clarify, and the desired number of centrifugations was recorded. After using Sevag reagent (isoamyl alcohol: chloroform = 1:4, *v/v*) to deproteinize, the crude PAPS was obtained by ethanol precipitation (1:2, *v/v*), dried, and weighed. The PAPS sample was dissolved to 10 g/L, and its rheological properties were observed. The PAPS yield was calculated as follows [6]:PAPS_1_ = the dried weight at extraction
PAPS_0_ = the dried weight at inoculation
$$\mathrm{PAPS}\;\mathrm{yield}(g/L)=\frac{{\mathrm{PAPS}}_{1}{-\mathrm{PAPS}}_{0}}{\mathrm{Volume}\;\mathrm{of}\;\mathrm{fermentation}\;\mathrm{broth}\;\mathrm{used}\;\mathrm{for}\;\mathrm{extraction}}$$

### 3.4. Gel Permeation Chromatography

The crude PAPS sample was suspended in 1 mL Milli-Q water and then filtered through a 0.22 μm syringe filter. We performed gel permeation chromatography analysis with an Agilent 1260 HPLC system equipped with a refractive index detector (Agilent Technologies Inc., Santa Clara, CA, USA), and both the Ultrahydrogel™ 500 gel-filtration chromatography column (7.8 × 300 mm) and Ultrahydrogel™ linear gel-filtration chromatography column (7.8 × 300 mm) (Waters, Milford, MA, USA) were eluted with 0.1 M Na_2_SO_4_ solution at 35 °C at a flow rate of 1 mL/min [60].

### 3.5. Monosaccharide Composition Analysis

Monosaccharide composition analysis was performed with the modified pre-column PMP derivative of carbohydrates [61]. Briefly, the sample (5 mg) was mixed with 4 mL of 2 mol/L TFA in a sealed tube. The mixture was hydrolyzed at 120 °C for 2 h. The excess TFA was removed by evaporating with methanol under reduced pressure 5 times. The lyophilized hydrolysate was then derivatized with 500 μL of 0.3 mmol/L NaOH and 500 μL PMP (0.5 mmol/L in methanol) at 100 °C for 60 min. After this, the reaction was terminated with 500 μL of 0.3 mmol/L HCl, and the system was supplemented to 2 mL by adding 500 μL of ultra-pure water. The reaction solution was extracted with 2 mL chloroform 3–5 times and the aqueous phase was retained. The derivatives were analyzed using a Shimazu HPLC system (Shimadzu Co, Tokyo, Japan) with a C18 column (4.6 mm × 250 mm, 5 μm). The UV detection wavelength was 250 nm, the mobile phase was 83% (*v/v*) sodium phosphate (0.1 mmol/L, pH 7.0) and 17% (*v/v*) acetonitrile, and the flow rate was 1 mL/min at 30 °C. The composition and content of PAPS were determined by comparing the retention time and peak area of standard monosaccharides.

### 3.6. Fourier-Transform Infrared Analysis

The Fourier-transform infrared (FT-IR) spectrum was used to evaluate the organic functional groups. An amount of 2 mg of crude PAPS powder was blended with 200 mg of potassium bromide (KBr) powder and pressed into a transparent film for infrared spectroscopy in the frequency range of 4000–400 cm^−1^.

### 3.7. Rheological and Mechanical Properties

PAPS samples were dissolved to a concentration of 1% (*v/w*) in Milli-Q water. The rheological measurements of PAPS samples were evaluated using an oscillatory rheometer (TA rheometer, DHR-1, USA) equipped with a 25 mm parallel plate at the proper gap. G’ and G’’ were determined by the frequency-sweep method. Frequency sweeps were performed at a constant 1% strain at 37 °C [62].

### 3.8. Determination of Soil’s Water-Stable Aggregates

Soil’s water-stable aggregates were isolated by a wet-sieving method with some modifications [63]. The 10% (*v/w*) crude PAPS (10 g/L) was added to the dry soil, while Milli-Q water was added to supplement the soil’s water content to 50%. The soil was then incubated at 25 ℃ for 30 days. Milli-Q water was added every three days to keep the soil’s moisture at 50% during the period. The soil was placed onto a sieve (height, 70 mm; diameter, 60 mm; mesh size, 200 mm), immersed in distilled water, and oscillated. The horizontal oscillations applied for 1 h were sinusoidal (2 cm amplitude, 98 oscillations min^−1^). The water-stable aggregates on the sieve (> 200 µm) were obtained, dried, and weighed.

### 3.9. Determination of Soil’s Water Evaporation Rate

The 10% (*v/w*) crude PAPS (10 g/L) was added to the dry soil, while Milli-Q water was added to supplement the soil’s water content to 50%. The soil was then incubated at 25 °C for 30 days. Milli-Q water was added every three days to keep the soil’s moisture at 50% during the period. Subsequently, the soil was placed in a constant temperature and humidity incubator (25 °C, 45% humidity). After 10 days of incubation, the soil was weighed, and the water evaporation rate was calculated.

### 3.10. Electron Microscopic Studies

We observed the microstructure of PAPS by scanning electron microscope (SEM). The dried samples were mounted on the sample holder and sputtered with silver powder and were observed with a Hitachi S4800SEM instrument at 15 kV.

The structure of the *P. alhagi* NX-11 was determined by SEM. Before observation, the bacteria collected by centrifugation (5000× *g*, 15 min) were fixed with 2.5% glutaraldehyde solution overnight. The immobilized bacteria were then dehydrated with graded concentrations of ethanol and finally freeze-dried for 48 h. The samples were then sputter-coated with gold prior to observation using a Hitachi S4800 instrument at 2 kV.

### 3.11. Extraction and Detection of Cell-Membrane Fatty Acids

After heating and centrifugation as described in Section 2.2, the bacteria were collected and were washed with 0.85% sterile NaCl for 3 times. The supernatant was discarded, and then 1.5 mL of methanol solution with a concentration of 1 mol/L sodium hydroxide was added to the 0.5 g bacterial solution; the mixture was shaken vigorously for 1.5 min and placed at 4 °C for 10 min. Fatty acid was extracted with 1 mL n-hexane solution by shaking for 1 min and then standing for 5 min, and n-hexane was collected by centrifugation (6000× *g* for 5 min). We filtered the solution with a 0.22 μm organic filter and placed the prepared sample in a gas-phase bottle to determine the fatty acid content. GC-MS was used to analyze the composition and content of fatty acids in the cell membrane. The capillary column was 0.25 mm × 30 m × 0.25 μm, the carrier gas was high-purity helium, and the injection volume was 1 μL (splitless injection). The injection port temperature was 240 °C, the column flow rate was 1 mL/min, and the initial column temperature was 160 °C. The temperature was increased to 190 °C at 5 °C/min for 5 min and then increased to 220 °C at 2 °C/min and held for 5 min.

### 3.12. Statistical Analysis

All experiments were performed three times, and the results are shown in average ± standard error (SE). Data were studied with analysis of variance (ANOVA); and Duncan went through multi-range testing with SPSS v.17.0 (SPSS, Chicago, IL, USA).

## 4. Discussion

In this study, we adequately explored the effect of changing pH conditions on the encapsulated structure to increase the exopolysaccharide extraction efficiency. We found that a pH level of 12 was most favorable for the extraction of PAPS derived from *P. alhagi* NX-11. The yield, molecular weight, and monosaccharide components of PAPS treated with pH 12 were not significantly changed compared with those treated with conventional pH (pH 7) (Figure 1, Figure 2 and Figure 3), which indicated that the structure of PAPS did not shift significantly, which was well proved by the FT-IR spectra and SEM (Figure 4 and Figure 8). Importantly, the pH−12-treated PAPS could be isolated after only two centrifugations, which considerably improved extraction efficiency compared with the original eight centrifugations (Figure 1B) [44]. In addition, pH−12-treated PAPS exhibited rheological properties analogous to pH−7-treated PAPS. We also found that PAPS treated with pH 12 condition had a relatively positive effect on the formation of soil aggregates and the evaporation of soil moisture (Figure 5, Figure 6 and Figure 7), which indicates that the physical properties of PAPS under the condition of pH 12 are better and have higher potential in practical application. Wang et al. found that during the later period of fermentation, cells of the engineered *C. glutamicum* strain were encapsulated following HA accumulation and limited the release of HA. We observed that the pH−12-treated PAPS had a significantly reduced encapsulated structure compared with those treated at pH 7, and the proportion of unsaturated fatty acids in pH−12-treated PAPS was higher than in pH−7-treated PAPS, which also indicated that the cell-membrane fluidity of samples treated at pH 12 was better. We guess that this is conducive to the extraction of PAPS without destroying the structure of PAPS.

Overall, we found that a pH level of 12 was most conducive to the extraction of PAPS derived from *P. alhagi* NX-11 without destroying the structure of PAPS by reducing the encapsulated structure of PAPS with *P. alhagi*. The experimental results further improve the possibility of industrial production of PAPS from *P. alhagi* NX-11 and provide a reference for the extraction of exopolysaccharides from other microorganisms.

## Figures and Tables

**Figure 1 molecules-27-07209-f001:**
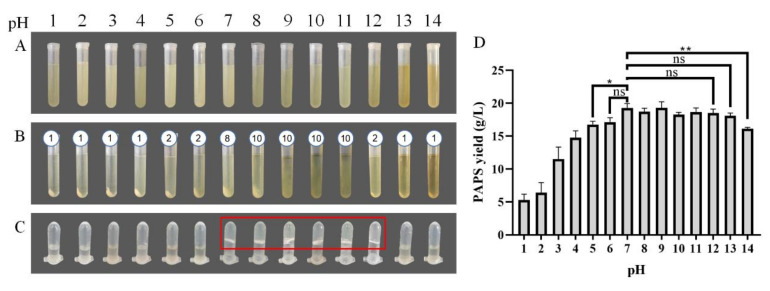
PAPS extraction process. (**A**) The fermentation broth was adjusted to the corresponding pH and heated. (**B**) The bacteria were separated by centrifugation. The number above the tubes represented the number of centrifugations required to obtain the supernatant. (**C**) The PAPS samples were dissolved to 10 g/L. (**D**) PAPS yield under different pH condition. The ns indicates no statistical significance of differences; * indicates the statistical significance of differences at *p* < 0.05; ** indicates the statistical significance of differences at *p* < 0.01.

**Figure 2 molecules-27-07209-f002:**
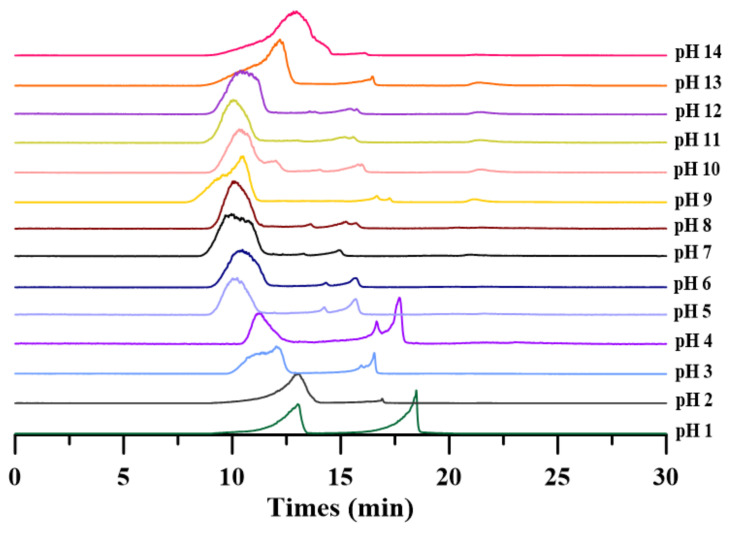
Crude PAPS size-fractionated by gel permeation chromatography.

**Figure 3 molecules-27-07209-f003:**
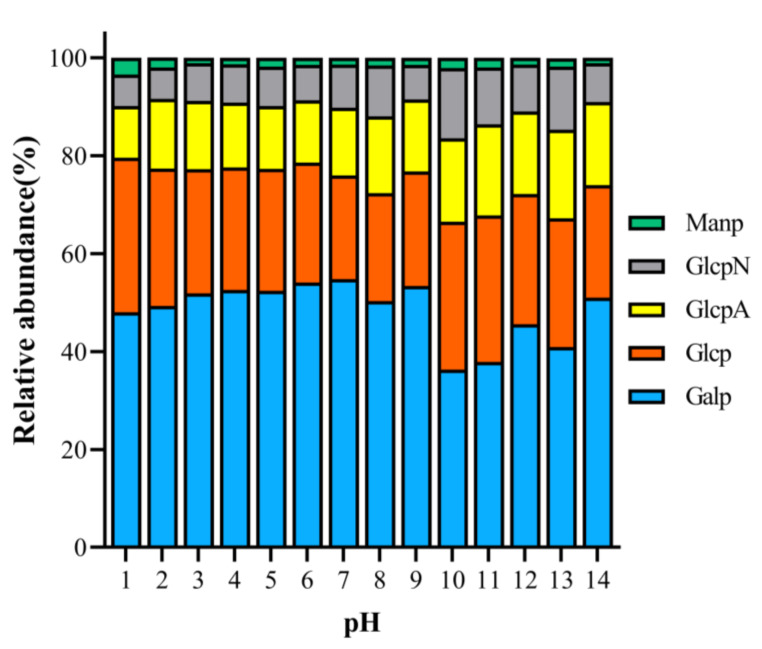
Analysis of monosaccharide components of pH−1–14-treated PAPS.

**Figure 4 molecules-27-07209-f004:**
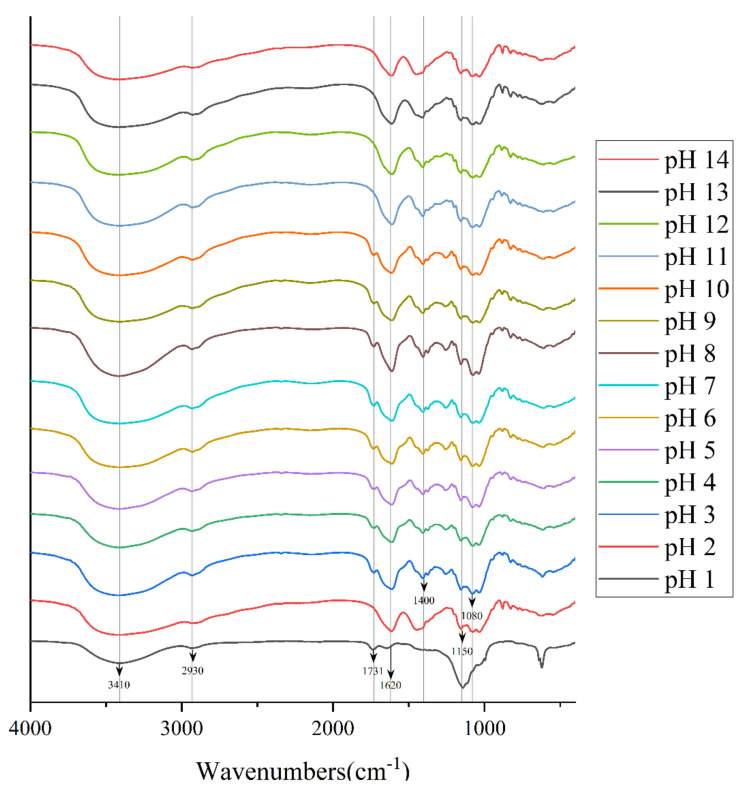
FT-IR spectrum of pH−1–14-treated PAPS samples.

**Figure 5 molecules-27-07209-f005:**
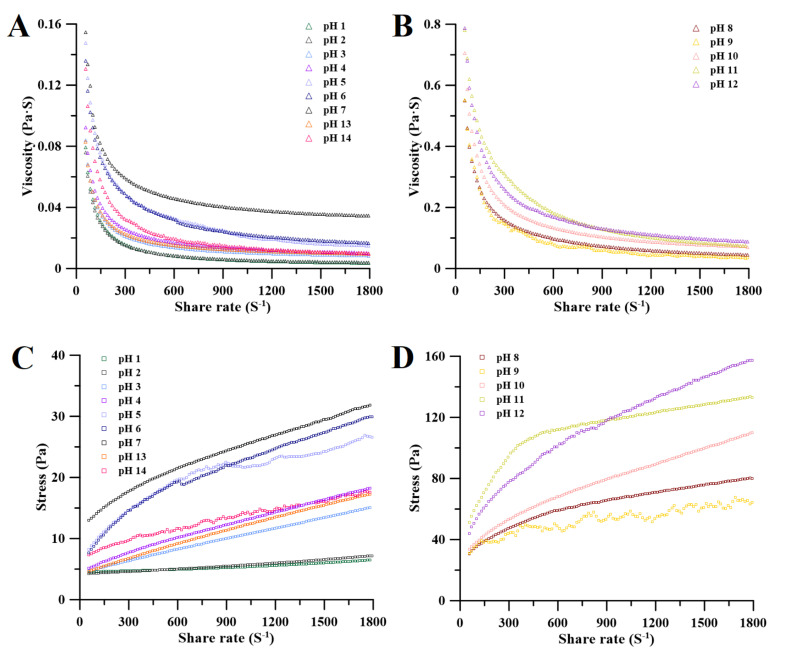
Viscosity (**A**,**B**) and stress (**C**,**D**) of pH−1–14-treated PAPS samples versus shear rate in aqueous solution (1% *w*/*v*).

**Figure 6 molecules-27-07209-f006:**
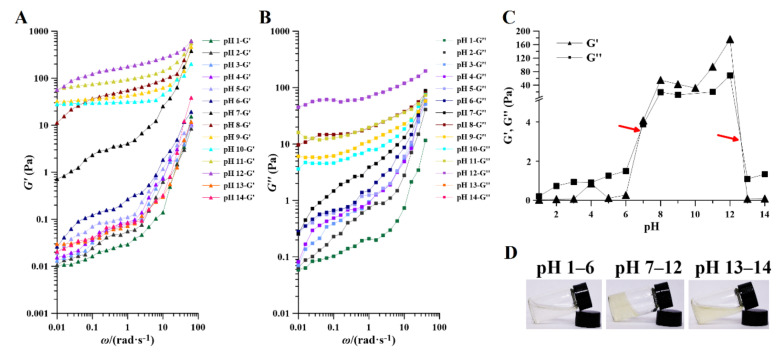
Effect of pH on the rheology of pH−1–14-treated PAPS. Storage (**A**) and loss (**B**) moduli and of pH−1–14-treated PAPS samples with angular frequency increasing at an aqueous solution (1% *w*/*v*). (**C**) The relationship between the modulus and pH of PAPS at an aqueous solution (1% *w*/*v*). (**D**) The apparent morphology of PAPS at an aqueous solution (1% *w*/*v*) under different pH.

**Figure 7 molecules-27-07209-f007:**
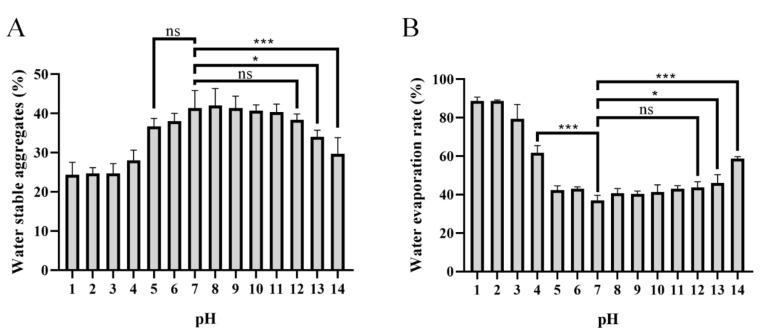
Effects of pH−1–14-treated PAPS on water-stable soil macro-aggregates (**A**) and water evaporation (**B**). The ns indicates no statistical significance of differences; * indicates the statistical significance of differences at *p* < 0.05; *** indicates the statistical significance of differences at *p* < 0.001.

**Figure 8 molecules-27-07209-f008:**
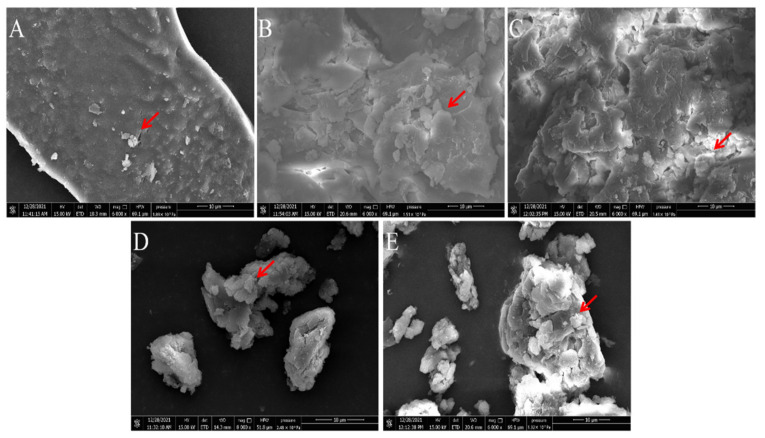
Comparison of molecular morphology of PAPS extracted by different methods ((**A**) under pH−2 conditions; (**B**) under pH−7 conditions; (**C**) under pH−9 conditions; (**D**) under pH−12 conditions; (**E**) under pH−13 conditions). Magnification is indicated in the micrographs.

**Figure 9 molecules-27-07209-f009:**
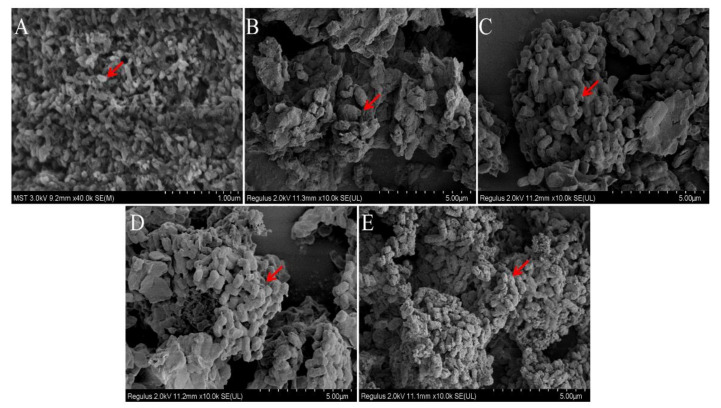
Comparison of molecular morphology of *P. alhagi* treated by different methods ((**A**) under pH−2 conditions; (**B**) under pH−7 conditions; (**C**) under pH−9 conditions; (**D**) under pH−12 conditions; (**E**) under pH−13 conditions). Magnification is indicated in the micrographs.

**Table 1 molecules-27-07209-t001:** Effects of initial pH value on fatty acid proportions in the membrane of *P. alhagi*.

Fatty Acid Species	pH 2 (%)	pH 7.0 (%)	pH 9 (%)	pH 12 (%)	pH 13 (%)
C8:0	0.50 ± 0.01 ^d^	7.53 ± 0.18 ^c^	5.21 ± 0.18 ^c^	11.89 ± 0.15 ^b^	32.09 ± 0.99 ^a^
C10:0	1.39 ± 0.04 ^d^	2.80 ± 0.17 ^b^	0.85 ± 0.02 ^d^	1.81 ± 0.04 ^c^	9.56 ± 0.43 ^a^
C11:0	0.04 ± 0.01 ^e^	0.84 ± 0.03 ^b^	0.17 ± 0.02 ^d^	0.37 ± 0.01 ^c^	2.08 ± 0.14 ^a^
C12:0	17.71 ± 0.57 ^b^	8.72 ± 0.11 ^c^	8.26 ± 0.03 ^c^	11.36 ± 0.06 ^a^	2.65 ± 0.12 ^d^
C13:0	0.07 ± 0.01 ^e^	0.48 ± 0.01 ^d^	0.79 ± 0.01 ^b^	0.67 ± 0.01 ^c^	0.99 ± 0.02 ^a^
C14:1	0.52 ± 0.01 ^d^	0.70 ± 0.02 ^d^	1.46 ± 0.10 ^c^	2.42 ± 0.04 ^b^	5.35 ± 0.16 ^a^
C14:0	3.82 ± 0.24 ^c^	4.44 ± 0.05 ^c^	4.92 ± 0.05 ^c^	7.14 ± 0.05 ^b^	14.48 ± 0.84 ^a^
C15:0	0.30 ± 0.01 ^d^	1.02 ± 0.03 ^b^	0.52 ± 0.01 ^c^	0.87 ± 0.01 ^c^	2.32 ± 0.14 ^a^
C16:1	8.72 ± 0.32 ^d^	17.6 ± 0.09 ^b^	16.73 ± 0.08 ^c^	22.25 ± 0.02 ^a^	2.42 ± 0.13 ^e^
C16:0	22.62 ± 0.5 ^b^	25.45 ± 0.10 ^a^	26.69 ± 1.90 ^a^	0.03 ± 0.01 ^d^	3.92 ± 0.17 ^c^
C17:1	0.43 ± 0.01 ^d^	0.65 ± 0.02 ^c^	0.17 ± 0.01 ^e^	0.80 ± 0.01 ^b^	2.16 ± 0.14 ^a^
C17:0	0.38 ± 0.01 ^c^	0.29 ± 0.02 ^c^	0.42 ± 0.02 ^c^	0.68 ± 0.01 ^b^	2.41 ± 0.11 ^a^
C18:3	0.25 ± 0.02 ^c^	0.79 ± 0.01 ^a^	0.43 ± 0.01 ^b^	0.75 ± 0.01 ^a^	0.78 ± 0.02 ^a^
C18:2	0.18 ± 0.01 ^e^	0.53 ± 0.01 ^c^	0.41 ± 0.01 ^d^	1.06 ± 0.02 ^b^	3.01 ± 0.03 ^a^
C18:1	22.37 ± 0.29 ^c^	21.25 ± 0.14 ^d^	25.3 ± 0.05 ^b^	29.43 ± 0.07 ^a^	0.99 ± 0.04 ^e^
C18:0	18.86 ± 0.47 ^a^	2.81 ± 0.03 ^d^	2.83 ± 0.13 ^d^	4.63 ± 0.02 ^c^	7.18 ± 0.28 ^b^
C20:4	0.22 ± 0.01 ^c^	0.59 ± 0.05 ^b^	0.27 ± 0.02 ^c^	0.63 ± 0.01 ^b^	2.57 ± 0.10 ^a^
C20:5	0.11 ± 0.01 ^c^	0.61 ± 0.02 ^b^	0.27 ± 0.01 ^c^	0.59 ± 0.01 ^b^	1.33 ± 0.05 ^a^
C20:3	0.28 ± 0.02 ^d^	1.49 ± 0.08 ^a^	0.27 ± 0.01 ^d^	0.80 ± 0.01 ^c^	1.09 ± 0.01 ^b^
C20:2	0.38 ± 0.03 ^e^	0.85 ± 0.06 ^c^	0.64 ± 0.02 ^d^	0.97 ± 0.01 ^a^	0.90 ± 0.03 ^b^
C20:1	0.15 ± 0.01 ^d^	0.22 ± 0.02 ^d^	0.32 ± 0.01 ^c^	0.53 ± 0.01 ^b^	0.73 ± 0.03 ^a^
U/S	0.51 ± 1.01 ^d^	0.83 ± 0.01 ^c^	0.91 ± 0.04 ^b^	1.53 ± 0.01 ^a^	0.27 ± 0.01 ^e^

^a^ Numbers are mean ± SD (*n* = 3). Different superscript letters indicate significant differences in the average of all samples (*p* < 0.05) within the same row. U/S: the ratio of unsaturated to saturated fatty acids.

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
