# Peer review of "High-Efficiency Extraction of Pantoea alhagi Exopolysaccharides Driven by pH-Related Changes in the Envelope Structure"

_molecules, 2022, doi:10.3390/molecules27217209_

Round 1

Reviewer 1 Report

Abstract: Good

Keywords: Add the species name

Line 29 -31: This is confusing, exopolysaccharides are polysaccharide. They are exo (obtain from extracelullar) and endo (obtain from intracellular). Kindly revise

First paragraph: How do you compare the stability with fungal exopolysaccharides?

Results: Figure 4, please combine them in one big to clearly visualize the peak differences. Please refer https://www.sciencedirect.com/science/article/abs/pii/S1878818119312216

Please provide the picture of dried EPS. 

Please describe the PA fermentation in detail. Kindly refer https://www.ncbi.nlm.nih.gov/pmc/articles/PMC7755588/

What was the fermentation system to grow the PA cell? shake flask or bioreactor system? If so, please provide clear fermentation calculations.

Why did you stop at FTIR and not NMR?

Discussion is weak, what was the relationship of MW with the yield and monosaccharide?

Additional comments:

1. picture of PA / macro and micromorphology

2. Graphical abstract 

Author Response

Dear Reviewer:

Thank you very much for your comments. We would like to express appreciations to you for suggestions on our manuscript entitled “High-efficiency extraction of Pantoea alhagi exopolysaccharides driven by pH-related changes in the envelope structure” (molecules-1959199). We have tried our best to revise the manuscript for your comments. We hope the revised paper will satisfy you.

Q1: Keywords: Add the species name.

According to your suggestion, we have Added the species name in Keywords. line 24

Q2: Line 29 -31: This is confusing, exopolysaccharides are polysaccharide. They are exo (obtain from extracelullar) and endo (obtain from intracellular). Kindly revise

According to your suggestion, we have revised the problem in line 27-29.

Q3: First paragraph: How do you compare the stability with fungal exopolysaccharides?

In response to your query, we have revised lines 33-37 of the first paragraph:

Moreover, most bacterial exopolysaccharides are more stable in production than fungal exopolysaccharides, It was reported that Ganoderma lucidum production of polysaccharide is directly affected by the morphology of mycelial pellet. Small-loosely branched myceli-um pellets produce higher polysaccharide compared to large mycelium pellets(AIMS Microbiol 2020, 6, 379-400).

Q4: Results: Figure 4, please combine them in one big to clearly visualize the peak differences.

Thanks for your suggestion, we have redrawn Figure 4.line184

Q5: Please provide the picture of dried EPS.

Thanks for your suggestion, we have provided the picture of dried EPS in Figure. S1.

Q6: Please describe the PA fermentation in detail

Thanks for your suggestion, we added the description of P. alhagi fermentation in 3.2. line356-363

Q7: What was the fermentation system to grow the PA cell? shake flask or bioreactor system? If so, please provide clear fermentation calculations.

In response to your query, we added the description of P. alhagi fermentation in 3.2 and provided fermentation calculations. line356-363, line374-376

Q8: Why did you stop at FTIR and not NMR?

Since NMR analysis is a heavy job, we stop at FT-IR, NMR will be analyzed in subsequent studies.

Q9: Discussion is weak, what was the relationship of MW with the yield and monosaccharide?

With the decrease of MV, the yield showed a downward trend, which may be because the partial hydrolysis of PAPS under acid or base conditions led to the decrease of the extraction yield of PAPS. The relationship between molecular weight and MV is unknown.

Q10: Picture of PA / macro and micromorphology

Thanks for your suggestion, we added the Picture of P. alhagi in Figure. S2.

Q11: Graphical abstract

Thanks for your suggestion, we will add the Graphical abstract.

We are looking forward to hearing from you soon.

Sincerely,

Peng Lei.

Reviewer 2 Report

1. What is the rationale of performing this study. How this study different from the previously performed study.

2. abstract must be revised with important findings

3. Why there is 2 peaks at low pH.

4. It is tough to interprete the IR graph. Use overlay plot with marking of important peaks.

5. Discussion must be elaborated. The findings must be supported with extensive discussion.

Author Response

Dear Reviewer:

Thank you very much for your comments. We would like to express appreciations to you for suggestions on our manuscript entitled “High-efficiency extraction of Pantoea alhagi exopolysaccharides driven by pH-related changes in the envelope structure” (molecules-1959199). We have tried our best to revise the manuscript for your comments. We hope the revised paper will satisfy you.

Q1: What is the rationale of performing this study. How this study different from the previously performed study.

The rationale of performing this study is that the packaging structure of PAPS and P. alhagi is affected after treatment under different pH conditions to improve the extraction efficiency of PAPS. In previously performed studies, researchers adjusting pH to increase the extraction efficiency of exopolysaccharides, which is generally believed to be based on the influence of polysaccharide molecular weight change on the rheological properties of solution to improving the extraction efficiency of polysaccharide.

Q2: Abstract must be revised with important findings.

According to your suggestion, we have revised the abstract in line 10-23 .

Q3: Why there is 2 peaks at low pH.

In response to your query, we have changed lines 131-133 in 2.2:

At low pH (pH 1-4), the molecular weight of PAPS was divided into two parts, we suspect that this is resulted in partial acid hydrolysis of PAPS, both of which were significantly lower than that of PAPS extracted at pH 7.

Q4: It is tough to interprete the IR graph. Use overlay plot with marking of important peaks.

Thanks for your suggestion, we have redrawn Figure 4.line184

Q5: Discussion must be elaborated. The findings must be supported with extensive discussion.

Thanks for your suggestion, we have rewritten the discussion.line456-482

We are looking forward to hearing from you soon.

Sincerely,

Peng Lei.

Round 2

Reviewer 1 Report

Dear author

Good feedbacks

Figure 8 and 9. Comparison of molecular morphology of P. alhagi treated by different methods is fantastic and deserve more observation. You can try to add arrows as Figure 4 from this example https://www.nature.com/articles/s41598-019-52493-y, which will boost the SEM picture comparison.

I also noticed spelling errors in the manuscript, kindly check

myce-lium = mycelium

p. alhgi = p. alhagi

Author Response

Dear Reviewer:

Thank you very much for your comments. We would like to express appreciations to you for suggestions on our manuscript entitled “High-efficiency extraction of Pantoea alhagi exopolysaccha-rides driven by pH-related changes in the envelope structure” (molecules-1959199). We have tried our best to revise the manuscript for your comments. We hope the revised paper will satisfy you.

Q1: Figure 8 and 9. Comparison of molecular morphology of P. alhagi treated by different methods is fantastic and deserve more observation. You can try to add arrows as Figure 4.

Thanks for your suggestion, we have redrawn Figure 8,, Figure. 9.line 314,line 328.

Q2: I also noticed spelling errors in the manuscript, kindly check

myce-lium = mycelium p. alhgi = p. alhagi

According to your suggestion, we have revised the ‘P. alhgi’ to ‘P. alhagi’.

The '-' in 'myce-lium' is generated automatically at line breaks in MDPI format, which is unavoidable.

We are looking forward to hearing from you soon.

Sincerely,

Peng Lei.

Reviewer 2 Report

Accept

Author Response

Dear Reviewer:

Thank you for your recognition of our work. We will continue to work hard. 

Sincerely,

Peng Lei.